# Left Ventricular Hypertrabeculation (LVHT) in Athletes: A Negligible Finding?

**DOI:** 10.3390/medicina61010032

**Published:** 2024-12-28

**Authors:** Rokas Jagminas, Rokas Šerpytis, Pranas Šerpytis, Sigita Glaveckaitė

**Affiliations:** 1Faculty of Medicine, Vilnius University, LT-03225 Vilnius, Lithuania; 2Clinic of Cardiac and Vascular Diseases, Institute of Clinical Medicine, Faculty of Medicine, Vilnius University, LT-03225 Vilnius, Lithuania; rokas.serpytis@santa.lt (R.Š.); sigita.glaveckaite@santa.lt (S.G.)

**Keywords:** left ventricular hypertrabeculation, athlete’s heart, sports medicine, cardiac magnetic resonance imaging, non-compaction

## Abstract

Left ventricular hypertrabeculation (LVHT) used to be a rare phenotypic trait. With advances in diagnostic imaging techniques, LVHT is being recognised in an increasing number of people. The scientific data show the possibility of the overdiagnosis of this cardiomyopathy in a population of people who have very high levels of physical activity. We describe the case of a young athlete with no medical history, who presented with syncope during a marathon running race. Initial evaluation showed elevated troponin I; transthoracic echocardiography showed a trabeculated ventricle and subsequent cardiac magnetic resonance (CMR) revealed left ventricular hypertrabeculation (LVHT). During subsequent evaluation by tilt table testing, vasovagal syncope was identified as the likely aetiology of the syncope. The patient was advised to cease sports and stimulants like caffeine use. At the 29-month follow-up, CMR showed the normalisation of the non-compacted to compacted myocardial ratio and an improvement in left ventricular function, with no further syncopal episodes reported. This is an example of the physiological hypertrabeculation of the LV apex in a recreational endurance athlete, with the normalisation of the non-compacted to compacted myocardial layer ratio after detraining. Physiological hypertrabeculation, a benign component of exercise-induced cardiac remodelling, must be differentiated from non-compaction cardiomyopathy and other pathologies causing syncope. This case underscores the importance of distinguishing physiological hypertrabeculation from pathological LVHT in athletes, highlighting that exercise-induced cardiac remodelling can normalise with detraining.

## 1. Introduction

Left ventricular hypertrabeculation (LVHT) is characterised by the exaggerated prominence of the trabeculae and deep recesses within the left ventricle [1]. While the American Heart Association previously classified it as a genetic cardiomyopathy [2], recent guidelines from the European Society of Cardiology (ESC) now consider LVHT as a phenotypic trait rather than a distinct cardiomyopathy [3,4]. Due to the absence of morphometric evidence for ventricular compaction in humans [5,6], the term “hypertrabeculation” is preferred, especially when the condition is transient or appears in adulthood. This shift reflects the evolving understanding of this condition and acknowledges that it can occur in various contexts, including alongside other cardiac abnormalities. 

The clinical presentation of LVHT is diverse, ranging from asymptomatic individuals to those experiencing serious complications such as arrhythmias, thromboembolism, heart failure, and sudden cardiac death. Currently, there are no universally accepted diagnostic criteria for LVHT. The existing imaging-based criteria have been criticised for potentially overdiagnosing the condition, particularly in low-risk populations and athletes [7,8].

A reversible phenotype of excessive trabeculation has been reported in athletes [9]. This is recognised as a morphologic epiphenomenon related to the high cardiac preload demand associated with intensive physical exercise [10]. The prevalence of ratios fulfilling the excessive trabeculation criteria among competitive athletes by echocardiography ranges between 1.4% and 8.1% [11]. In the Progression of Early Subclinical Atherosclerosis (PESA) study, the incidence of isolated excessive trabeculation in a younger population (mean age 48 years) was doubled in those representing the highest quintile, achieving vigorous physical activity, compared to those with no vigorous physical activity [12]. The latter association is not consistent across studies. In the community-based UK Biobank study, there was no evidence to suggest a dose–response relationship between the physical activity intensity and the extent of left ventricular trabeculation [13]. However, when considering the available data, no adverse cardiac events have been reported in individuals with normal left ventricular function, regardless of the extent of left ventricular trabeculation [14]. It is essential to differentiate physiological LVHT from pathological LVHT (also called left ventricular non-compaction LVNC), which is associated with significant clinical implications and often requires intervention and physical exercise restriction.

In this case report, we present a clinical case of physiological LVHT in an athlete and discuss the contemporary knowledge and current controversies regarding physiological and pathological LVHT.

## 2. Case Presentation

A 21-year-old male, with no prior medical history, was hospitalised after a syncope that occurred during a marathon race. He lost consciousness after running 8 kilometres. At the time, the patient did not take any medications regularly, but, during adolescence, he took various supplements (carnitine, amino acids) while exercising. He denied any recent viral illness. There was no family history of premature cardiovascular death. The physical examination on admission was within normal limits, while the electrocardiogram (ECG) showed a partial right bundle branch block. Additionally, blood tests indicated elevated cardiac biomarkers (troponin I, creatine kinase MB fraction, B-type natriuretic peptide, myoglobin), which suggested myocyte and muscle injury as a consequence of exercise-induced stress (Table 1).

Initially, transthoracic echocardiography was performed (Appendix A). It showed the prolapse of both mitral valve cusps with mild mitral insufficiency. Additionally, 1st-degree left atrial dilatation and a non-dilated left ventricle (LV) with a normal ejection fraction of around 61% (as calculated by biplane Simpson) with the increased trabeculation of both the left and right ventricles was noticed.

To exclude coronary artery pathology or anomalies, computed tomography angiography (CTA) was performed (Appendix A). No changes in the coronary vessels were found. Cardiac magnetic resonance (CMR) was performed to assess structural and functional cardiac changes and to exclude cardiomyopathies. The latter test showed slight LV dilatation (left ventricular end-diastolic volume index (EDVI) 104 mL/m^2^ (normal range <92), left ventricular end-systolic volume index (ESVI) 47 mL/m^2^ (normal range <30), borderline LV systolic function (55%, normal range >55%)) and signs of LV hypertrabeculation. The trabeculated to compacted myocardium ratio was up to 2.6 in end-diastole (Appendix A, Figure 1A–C). According to Petersen’s criteria, a normal ratio of trabeculated to compacted myocardium is <2.3 in end-diastole in long-axis heart views [15]. A non-specific focus of fibrosis was also detected on late gadolinium enhancement (LGE) images at the inferior left and right ventricle insertion points (Figure 1C, white arrow).

After a comprehensive review of the patient’s medical history, and due to the lack of any registered arrhythmias during the continuous telemonitoring of the patient and while performing exercise electrocardiography, it was considered that the patient experienced vasovagal syncope during a marathon run. As a result, a tilt table test was performed, which revealed a mixed-type vasovagal syncope. Additionally, 24 h Holter monitoring was performed and showed a sinus rhythm with a heart rate (HR) of 28–91 beats per minute (bpm). Short bradycardia episodes with HR less than 25 bpm were observed. The average daily HR was 41 bpm.

At that time, it was concluded that the patient may have physiological remodelling of the LV due to his high activity in endurance sports. Moreover, it was strongly recommended to discontinue the use of various stimulants such as caffeine and reduce his physical activity from high to low–moderate in order to reduce the incidence of syncope. No cardiovascular medications were prescribed. At the time, the diagnosis was considered to represent pathological LVHT, warranting close monitoring and necessitating restrictions of physical activity to mitigate the risk of adverse outcomes.

The patient initially found the recommended reduction in physical activity challenging. However, with the support of his healthcare team, he successfully transitioned to a low, moderate-intensity exercise regimen. The patient embraced a holistic approach to well-being, incorporating mindfulness practices like yoga and meditation into his routine. These changes improved his physical health and enhanced his overall quality of life, allowing him to enjoy a balanced and fulfilling lifestyle.

After 2 years of detraining, the patient underwent a follow-up examination and additional CMR, which revealed the reverse remodelling of the LV with a decrease in left ventricular dilatation (EDVI from 104 to 97 mL/m^2^ (normal range < 92), ESVI from 47 to 40 mL/m^2^ (normal range < 30)); the LV ejection fraction at follow-up was within the normal range (58%) and the LV myocardium had a normal trabeculated to compacted myocardium ratio in diastole of 1.9 (normal range < 2.3) (Figure 1D–F, Appendix A). Additionally, a laboratory work-up showed a normal value of BNP < 10 ng/L with complete normalisation of transaminase and electrolytes. The patient reported no recurrence of syncopal episodes.

## 3. Discussion

LVHT is a complex and under-researched heart condition. The diagnosis of LVHT has increased significantly due to advancements in imaging techniques, yet there remains considerable debate over its diagnostic criteria and management [15,16]. Recent studies have shown that LVHT, previously underestimated in prevalence, is more common than thought, affecting 0.14% to 0.27% of the general population and 9.5% of children with cardiomyopathies [16]. The condition is diagnosed more frequently due to improved awareness and imaging technology. Isolated LVHT has been found in up to 8% of athletes, suggesting a potential overlap with physiological adaptation in this group [16].

The prognosis of LVHT remains uncertain, despite it being recognised as a clinical condition for over 30 years. Recent studies indicate that the overall survival is lower in patients with LVHT compared to the expected survival of age- and sex-matched individuals from the general U.S. population. However, patients with a preserved left ventricular ejection fraction and isolated apical non-compaction have a survival rate similar to that of the general population [17,18].

In 2020, the European Society of Cardiology (ESC) proposed guidelines for recreational and professional athletes diagnosed with LVHT [14]. According to them, participation in high-intensity exercise and all competitive sports, if desired, with the exception of instances where syncope may cause serious harm or death, may be considered (IIbC) in asymptomatic individuals with LVHT and LVEF ≥50% and the absence of frequent and/or complex ventricular arrhythmias (VAs). However, participation in recreational exercise programmes of low to moderate intensity may be considered (IIbC) in individuals with LVEF 40–49% in the absence of syncope and frequent or complex VAs on ambulatory Holter monitoring or exercise testing. Patients presenting with a gene positive for LVHT but phenotype-negative (except lamin A/C or filamin C carriers) may be considered (IIbC) for high- or very high-intensity sports. Participation in high-intensity exercise or competitive sports is not recommended (IIIC) in individuals with any of the following: symptoms, LVEF < 40% and/or frequent and/or complex VAs on ambulatory Holter monitoring or exercise testing. Finally, an annual check-up for risk stratification is recommended for recreational athletes with genotype-positive/phenotype-negative LVHT [14].

Athletes are generally healthy but might have certain cardiac disorders, which might, during athletic participation or training, result in cardiac symptoms including syncope. Vasovagal syncope is probably the most common cause of syncope in athletes, but syncope in the context of these cardiac disorders might be a precursor to sudden death. Consequently, the evaluation of syncope assumes extreme importance, particularly if these underlying cardiac conditions, such as cardiomyopathy, myocarditis, channelopathies (long QT, Brugada syndrome, etc.), pre-excitation syndrome, and coronary artery anomalies or disease, are present [19]. In our patient, the latter cardiac conditions were excluded, making vasovagal syncope the most probable reason for the loss of consciousness during the marathon run.

For athletes, activity restrictions based on uncertain diagnoses can have a devastating psychological impact. The fear of sudden cardiac death can be paralyzing, leading to anxiety, depression, and a loss of self-esteem. The inability to participate in their chosen sport can erode their sense of identity and purpose. This can be particularly challenging for young athletes who are still developing their sense of self and navigating the complexities of adolescence. To address these challenges, access to mental health support is crucial. Psychotherapy or sports psychology can provide athletes with valuable coping mechanisms, such as cognitive behavioural therapy techniques to manage anxiety and develop a more positive mindset.

A recent study involving 1492 Olympic elite athletes across various sports disciplines who underwent electrocardiograms, echocardiograms, and exercise stress tests found that LVHT was common, occurring in 29% of participants, particularly in male, Afro-Caribbean, and endurance athletes [20]. LVTs in this population were interpreted as a manifestation of adaptive remodelling associated with elite athletic training. The study concluded that, in the absence of clinical abnormalities such as LV systolic or diastolic impairment, electrocardiographic repolarisation abnormalities, or a family history of cardiomyopathy, LVHT in athletes is of benign clinical significance and does not require further investigation. In addition, this prospective study suggested that recreational marathon running does not increase left ventricular trabeculation [21]. However, further investigations are needed to confirm this hypothesis.

Applying cut-off values from published LVHT criteria to young, healthy individuals creates a potential risk for overdiagnosis. It has previously been shown that younger individuals possess greater amounts of apical trabeculation [22,23], but age-specific normative or cut-off values for pathological LV trabeculation do not currently exist. In this sample of healthy subjects, excessive trabeculation was found predominantly at the LV apex, which is recognised as the most commonly non-compacted segment [15] and was detected with greater sensitivity by Chin [24] and Captur [25] using apical fractal dimension (FD) criteria. A higher prevalence of positive Chin [26] as compared to Jenni [11] criteria has previously been reported in Olympic athletes with prominent trabeculation [27]. The fractal dimension (FD) quantifies how thoroughly a complex structure fills a space. Its value is constrained by the structure’s topological dimension. For example, in two-dimensional imaging, endocardial borders are more complex than straight lines, giving them an FD greater than 1. However, since they do not completely occupy the two-dimensional space, their FD remains below 2. Thus, the FD for an endocardial border consistently falls between 1 and 2, representing a non-integral value. In LVHT, excessive trabeculations result in a highly irregular endocardial border. Fractal analysis of these borders in LVHT is expected to produce a higher FD compared to normal hearts. Thus, the ESC recommends the consideration of (IIaB) pathologic LVHT (also called left ventricular non-compaction (LVNC)) in athletic individuals if they fulfil the imaging criteria, in association with cardiac symptoms, a family history of LVNC or cardiomyopathy, LV systolic (EF < 50%) or diastolic (E’ < 9 cm/s) dysfunction, a thin compacted epicardial layer (<5 mm in end-diastole on CMR or <8 mm in systole on echocardiography), or abnormal 12-lead ECG [14].

Studies in athletes have reported prominent trabeculation, raising concerns about a diagnostic grey zone between LVHT and exercise-induced remodelling [28]. Severe cases in clinical practice suggest that LVHT might be a distinct pathology, as the extent of trabeculation exceeds what could arise from adaptation alone. While the adult myocardium can remodel via myocyte hypertrophy, its limited proliferative capacity makes extensive de novo trabeculation unlikely. Most cases likely reflect normal trabeculation variations influenced by the ventricular geometry and loading conditions. Clinical evaluations, including symptoms, family history, function, arrhythmias, and CMR findings, suggest that only 0.1% of cases align with LVHT, with most being normal or non-cardiomyopathic. This supports the idea that prominent trabeculation in athletes is often exercise-induced remodelling.

In this case, insertion point fibrosis was detected on lGE images at the inferior RV insertion point (Figure 1C, white arrow). RV insertion point fibrosis is frequently observed in athletes irrespective of age [29,30]. Its prevalence has been reported in 20–30% of athletes and is linked to the combination of the training load and intensity [31]. One hypothesis suggests that this pattern may result from pressure or volume overload in the right ventricle during intense exercise, causing micro-injuries that manifest as late gadolinium enhancement (LGE) [32]. In healthy elderly individuals, insertion point fibrosis may represent a normal ageing process and is often deemed an incidental finding when unaccompanied by other signs of cardiac damage [33]. Generally, it is considered to be benign in structurally normal hearts [34,35]. A similar LGE pattern has been observed in congenital heart disease patients with RV volume and pressure overload, in non-ischaemic dilated cardiomyopathy. The RV insertion LGE in the latter disease entities is associated with more advanced LV remodelling and a higher risk of clinical events [36,37].

Increased cardiac troponin levels following exercise have been documented in athletes across various sports [38,39]. These increases are temporary and generally return to baseline within 48–72 h after exercise [40]. While the exact mechanisms behind exercise-induced troponin release remain unclear [41], recent research suggests that activities such as marathon running may impact the cardiomyocyte integrity [42], potentially leading to the leakage of cytosolic troponin fragments [43]. Further investigation is required to determine whether specific groups (based on factors such as age, sex, or sports discipline) with exercise-induced troponin elevations might be at a heightened cardiovascular risk [44].

Differentiating physiological and pathological left ventricular hypertrabeculation is a complex clinical challenge. While physiological LVHT is a normal adaptive response to an increased cardiac workload, pathological LVHT (LVNC) can be a indicator of an underlying heart disease, potentially leading to arrhythmias and sudden cardiac death. Both can present with similar echocardiographic findings, making it difficult to definitively classify them based on imaging alone. Additionally, there is a lack of standardised diagnostic criteria, further complicating the assessment [45].

Because LVHT often exhibits a familial pattern, the screening of relatives is recommended. Genetic testing can help to confirm the diagnosis and assess the future risk. Genetic variants associated with LVHT carry varying levels of risk for adverse outcomes [46]. High-risk variants, such as TTN (Titin) and complex genotypes involving multiple pathogenic mutations, are strongly linked to reduced LVEF, heart failure, and increased mortality. Moderate-risk variants, including MYH7 (Myosin Heavy Chain), ACTC1 (Alpha Cardiac Actin), and MYBPC3 (Myosin-Binding Protein C), are associated with ventricular arrhythmias and structural abnormalities but may confer a better prognosis compared to high-risk variants. Low-risk variants, often found in desmosomal or cytoskeletal genes, are less frequently linked to severe outcomes, although they still contribute to the LVHT phenotype. The overall risk depends on clinical factors such as the LVEF, LGE on CMR, and the presence of arrhythmias, emphasising the need for individualised assessment and management.

Unfortunately, there is no standardised screening protocol for patients presenting with LVHT. A policy proposed by Paluszkiewicz et al. highlights the importance of a patient’s symptoms at presentation, referring to a different diagnostic approach [44]. Asymptomatic individuals undergo a thorough evaluation, including a family history review and a 12-lead ECG. Depending on the initial findings, further assessment may involve echocardiography or CMR. Fibrosis and reduced LVEF are associated with a higher risk of complications. Patients with no significant abnormalities on these tests have a low risk of cardiac events and typically do not require genetic testing or extensive follow-up. However, it is crucial to consider the possibility of transient or adaptive hypertrabeculation, which may resolve without intervention. Symptomatic individuals require a comprehensive cardiac evaluation, including a 12-lead ECG, 24 h Holter monitoring, echocardiography, and CMR. A detailed three-generation family history is essential. All first-degree relatives should undergo a cardiac evaluation, including echocardiography and CMR. Coexisting conditions, such as neuromuscular disorders and congenital heart defects, warrant careful consideration. Genetic testing using panels for cardiomyopathies associated with LVHT is recommended for all symptomatic patients. Initial testing should be performed on the index patient within the family. If genetic abnormalities are detected, the family should be offered genetic counselling and testing [47].

Due to the morphological diversity of LVHT, various classification systems have been proposed [44]. In 2016, Arbustini et al. [45] introduced a seven-subtype classification (Table 2). Another common classification system proposed by Van Waning et al. [48] recognises four subtypes: isolated left ventricular hypertrabeculation (iLVHT), LVHT with hypertrophic cardiomyopathy (HCM), LVHT with dilated cardiomyopathy (DCM), and a combination of all three. Most studies, however, analyse LVNC as a single entity, making direct comparisons between different studies challenging.

Research indicates that young athletes diagnosed with conditions such as hypertrophic cardiomyopathy or long QT syndrome experience heightened psychological distress due to the life-altering nature of these diagnoses [49]. The uncertainty surrounding the diagnosis of LVHT—whether it is pathological or physiological—can exacerbate these feelings, as athletes may grapple with the fear of being sidelined from their sport, without a clear understanding of their condition’s implications [50]. The ethical responsibility of healthcare providers includes ensuring that athletes receive comprehensive support during the diagnostic process. This involves not only accurate diagnosis but also providing psychological support to help them cope with the potential implications of their diagnosis.

In this case, LV hypertrabeculation reached its maximum expression at the peak of physical conditioning and decreased two years after the detraining period. This suggests that LV hypertrabeculation occurred during the marathon training period as a response to the intensity and volume of the training load. These changes represent adaptive mechanisms that can regress during detraining.

Future studies aimed at distinguishing between physiological and pathological LVHT could greatly benefit from the integration of sophisticated imaging techniques, such as utilising fractal analysis in conjunction with CMR, the use of higher-spatial-resolution imaging to better delineate trabeculations and their relationship with myocardial function, or the integration of machine learning algorithms for the automated detection and quantification of trabeculation in CMR images. This could enhance the diagnostic accuracy and reduce the interobserver variability. Research should focus on developing and validating these algorithms against histological findings to ensure their reliability in clinical practice.

## 4. Conclusions

Isolated LVHT may be an adaptive mechanism for training in up to 8% of competitive athletes. After the detraining period, preload-related eccentric LV remodelling may normalise, and the trabeculations collapse at their base with the disappearance of the diagnostic imaging criteria of hypertrabeculation. From a clinical point of view, the presence of asymptomatic LVHT with normal LV function does not increase the risk of adverse cardiac events, but no data exist regarding the routine disqualification of these athletes from participation in high-intensity exercise and competitive sports. Athletes who either have LV systolic dysfunction or dilatation, cardiac symptoms, or abnormal electrocardiographic findings unrelated to training should undergo comprehensive cardiac examination. Recommendations on sports restrictions in these athletes are based on the presence of underlying cardiomyopathies and are not influenced by the additional presence of hypertrabeculation. There is a need for evidence-based high-quality data on the diagnostic criteria and prognostic impact of LVHT in athletes.

## Figures and Tables

**Figure 1 medicina-61-00032-f001:**
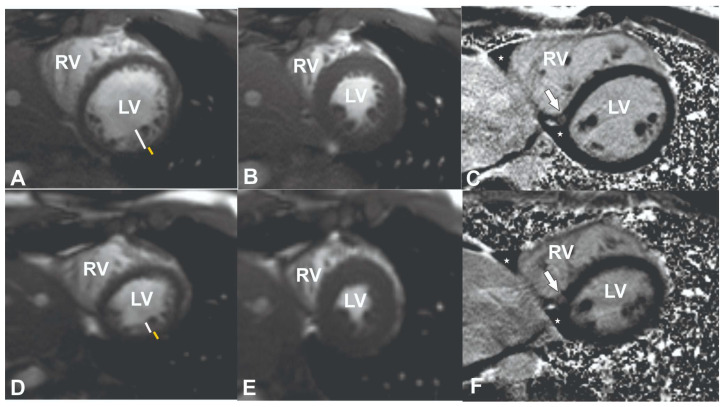
CMR images of the patient at admission (**A**–**C**) and follow-up (**D**–**F**). Cine short-axis images at the midventricular level at diastole (**A**) and systole (**B**), representing left ventricular hypertrabeculation (its thickness is denoted with a white line and the compact myocardial layer with a yellow line), respective midventricular slice in inversion recovery sequence 10–15 min after contrast media injection (late gadolinium enhancement sequence, arrow pointing to the fibrotic focus at the inferior left and right ventricle insertion point, asterisk represents traces of fluid in the pericardial space at the inferior cardiac wall). Follow-up cine short-axis images at the midventricular level at diastole (**A**) and systole (**B**), representing the normalisation of left ventricular hypertrabeculation and a decrease in the left ventricle volumes after detraining. On a late gadolinium enhancement image, the same findings as in (**C**) were noticed.

**Table 1 medicina-61-00032-t001:** Results of laboratory tests on admission and at follow-up. RV—reference value used in Vilnius University Hospital Santaros Clinics (VUL SK) laboratory, Vilnius, Lithuania. CK—creatine kinase, CK-MB—creatine kinase MB fraction isoenzyme, BNP—B-type natriuretic peptide, AST/GOT—aspartate aminotransferase/glutamate oxaloacetate transaminase, ALT/GPT—alanine aminotransferase/glutamate pyruvate transaminase, *—in acute heart failure, **—in chronic heart failure.

Laboratory Test	Results at Presentation	Follow-Up After 24 h	Follow-Up After 72 h	Reference Value (Male)
CK (U/L)	545.0	679.0	-	25.0–195.0
CK-MB (μg/L)	6.14	-	-	<5.2
Myoglobin, (μg/L)	455.9	-	-	<155.0
Troponin I (ng/L)	4456.4	2733.4	-	≤35.2
BNP (ng/L)	173.8	66.6	-	<100 */<35 **
D-dimers (µg/L)	500.0	-	-	<250.0
AST/GOT (U/L)	-	376.0	95.0	<40.0
ALT/GPT (U/L)	-	428.0	368.0	<40.0
Potassium (mmol/L)	4.3	4.5	5.0	3.8–5.3
Sodium (mmol/L)	140.0	140.0	141.0	134.0–145.0
Creatinine (μmol/L)	121.0	98.0	99.0	62.0–115.0

**Table 2 medicina-61-00032-t002:** Subtypes of LVHT, modified by Arbustini et al., 2016 [46]. iLVHT—isolated left ventricular hypertrabeculation, HCM—hypertrophic cardiomyopathy, DCM—dilated cardiomyopathy, RCM—restrictive cardiomyopathy, ARVC—arrhythmogenic right ventricular cardiomyopathy, HT—hypertrabeculation.

Classification of LVHT
1. iLVHT: HT morphology in LV with normal systolic and diastolic function, size, and wall thickness.
2. LVHT with LV dilation and dysfunction at onset, such as in the paradigmatic infantile CMP of Barth syndrome.
3. LVHT in hearts fulfilling the diagnostic criteria for DCM, HCM, RCM, or ARVC.
4. LVHT associated with congenital heart disease.
5. Syndromes with LVHT, either sporadic or familial, in which the non-compaction morphology is one of the cardiac traits associated with both monogenic defects and chromosomal anomalies, i.e., complex syndromes with several multiorgan defects.
6. Acquired and potentially reversible LVHT, which has been reported in athletes; it has also been reported in sickle cell anaemia, pregnancy, myopathies, and chronic renal failure.
7. RV non-compaction, concomitant with that of the LV or present as a unique anatomic area of HT.

## Data Availability

The data presented in this study are available on request from the corresponding author.

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
