# Peer review of "Left Ventricular Hypertrabeculation (LVHT) in Athletes: A Negligible Finding?"

_medicina, 2024, doi:10.3390/medicina61010032_

Round 1

Reviewer 1 Report

Comments and Suggestions for Authors

The topic is both novel and timely in the field of sports cardiology. It fills a unique gap in the literature by focusing on the need of proper diagnostic criteria in preventing unnecessary therapies for LVHT in athletes.

I have though of some suggestions:

1. To minimize misunderstanding among readers unfamiliar with the topic, the distinction between "pathological LVHT" and "physiological hypertrabeculation" may be made more explicitly at the start of the discussion.

2. Explain why less activity was indicated in this case, linking the results to broader clinical practices and guidelines.

3. Provide further information on the patient's adherence to recommendations (for example, how physical activity levels were lowered, psychological effects, and lifestyle improvements).

4. While the case report discusses improvement post-detraining, it does not go into detail about the patient's long-term prognosis or the risks of returning to high-intensity athletics, such as arrhythmias or progression to severe LV dysfunction.

5. Adding implications for sports physicians and cardiologists on screening and advising athletes with comparable findings might widen the report's significance.

6. It would be beneficial to briefly discuss future updates to diagnostic criteria or new approaches for differentiating physiological from pathological results.

7. Emphasize the ethical implications of limiting physical activity based on uncertain diagnoses and the potential psychological effects for athletes.

8. Include a more detailed discussion of:

How does this case fit into the wider context of athlete's heart vs. diseased LVHT?

Emerging diagnostic technologies and biomarkers can help distinguish between healthy and diseased situations.

Acknowledge the limits of single-case studies, such as their lack of generalizability.

Discuss the need for age- and activity-specific normative data for LV trabeculation to prevent overdiagnosis.

Suggestions for future study include the use of sophisticated imaging techniques (e.g., fractal analysis) to distinguish between normal and diseased trabeculation.

9. Include additional follow-up imaging data (if available) to illustrate the progression of cardiac remodelling over time because It has a lot of videos in the supplementary document, but only one image and one table in the case report  per se. 

Author Response

Thank you very much for your feedback. Please find our answer below.

1. To minimize misunderstanding among readers unfamiliar with the topic, the distinction between "pathological LVHT" and "physiological hypertrabeculation" may be made more explicitly at the start of the discussion.

A: An explanation was added in the introduction section (highlighted in yellow)..

2. Explain why less activity was indicated in this case, linking the results to broader clinical practices and guidelines.

A: The treatment strategy is explained in the discussion according to the 2015 AHA/ACC and 2020 ESC guidelines on sport activities.

3. Provide further information on the patient's adherence to recommendations (for example, how physical activity levels were lowered, psychological effects, and lifestyle improvements).

A: The patient’s adherence is added in the case presentation section.

4. While the case report discusses improvement post-detraining, it does not go into detail about the patient's long-term prognosis or the risks of returning to high-intensity athletics, such as arrhythmias or progression to severe LV dysfunction.

A: Excessive trabeculation in athletes is often a reversible morphological adaptation to high cardiac preload demands from intense physical activity. It is generally not associated with adverse outcomes if ventricular function is preserved and there are no other underlying cardiac abnormalities. In this case, it was a reversible hypertrabeculation phenomena, so the patient has a low risk of arrhythmias or progression to sever LV dysfunction. However, we added an additional paragraph in the discussion section about genetic variants associated with higher cardiovascular risks. In our case, we did not test our patient genetically.

5. Adding implications for sports physicians and cardiologists on screening and advising athletes with comparable findings might widen the report's significance.

A: We added recommendations to sports medicine doctors and cardiologists in the discussion section.

6. It would be beneficial to briefly discuss future updates to diagnostic criteria or new approaches for differentiating physiological from pathological results.

A: We added a paragraph talking about the demand for more studies on this topic.

7. Emphasize the ethical implications of limiting physical activity based on uncertain diagnoses and the potential psychological effects for athletes.

A: We added a paragraph talking about athletes’ potential psychological effects based on uncertain diagnoses.

8. Include a more detailed discussion of:

How does this case fit into the wider context of athlete's heart vs. diseased LVHT?

Emerging diagnostic technologies and biomarkers can help distinguish between healthy and diseased situations.

Acknowledge the limits of single-case studies, such as their lack of generalizability.

Discuss the need for age- and activity-specific normative data for LV trabeculation to prevent overdiagnosis.

A: Additional information was added in the discussion section to support our case report.

Suggestions for future study include using sophisticated imaging techniques (e.g., fractal analysis) to distinguish between normal and diseased trabeculation.

A: Thank you for your suggestion, but in our University hospital, we do not perform fractal analysis for clinical or research purposes. However, data on more sophisticated imaging techniques has been provided in the discussion section.

9. Include additional follow-up imaging data (if available) to illustrate the progression of cardiac remodelling over time because the supplementary document contains many videos, but the case report has only one image and one table.

A: We would like to notify you that Videos No. VIII, IX and X are CMR at 2 years follow-up. Also, due to the conditions of the journal and the size of video files, videos were merged into a .zip file and uploaded separately. Figure 1 represents baseline (panels A-C) and follow-up (panels D-F) images. Additionally, we have added data on laboratory biomarker dynamics in Case report section stating normalization of biomarkers (BNP at follow-up <10 ng/l).

Reviewer 2 Report

Comments and Suggestions for Authors

Peer Review for "Left Ventricular Hypertrabeculation (LVHT) In Athletes: Negligible Finding?"

Strengths of the Article

1. Clinical relevance:

   The article highlights an important clinical question regarding LVHT in athletes, a condition that can be easily over-diagnosed. The differentiation between pathological LVHT and exercise-induced cardiac remodeling is clinically significant.

2. Case Presentation clarity:  

   The case is well-documented, with a clear progression from presentation to diagnosis, treatment, and follow-up. The inclusion of imaging data and laboratory results provides a comprehensive picture.

3. Follow-Up data:

   The 2-year follow-up and demonstration of reverse remodeling after detraining is a key strength. This adds robust evidence to support the hypothesis of physiological versus pathological LVHT.

4. Literature integration:  

   The authors integrate recent ESC and AHA guidelines, as well as supporting studies, to provide a contemporary perspective on the diagnostic challenges and clinical significance of LVHT.

5. Visuals and supplementary data:  

   High-quality imaging (CMR and echocardiography) supplements the narrative effectively.

Weaknesses of the article

1. Limited scope:  

   The study focuses on a single case, which limits generalizability. Although the findings are interesting, a larger cohort study would provide stronger evidence.

2. Missing quantitative comparison:  

   The article could have benefited from a quantitative comparison with other athletes or individuals with LVHT to reinforce the argument of exercise-induced adaptation.

3. Ambiguity in syncope etiology:  

   While vasovagal syncope was diagnosed, the role of LVHT in syncope remains speculative. Clarifying this relationship would strengthen the clinical message.

4. Overreliance on imaging findings:  

   The study heavily emphasizes imaging results without adequate discussion on how clinical suspicion should guide the diagnostic process, particularly in asymptomatic athletes.

5. Writing style:

   Some sections, particularly in the discussion, are verbose and repetitive. Streamlining these parts would improve readability.

Overall assessment and Decision

The article addresses a niche yet clinically relevant topic. It is well-written and presents a clear case with valuable follow-up data. However, its conclusions would be more impactful if supported by larger studies or stronger quantitative analysis.

Decision: Minor revisions recommended.

Specific recommendations for Revision

1. Clarify syncope pathophysiology:  

   Clearly differentiate the role of LVHT from vasovagal syncope and explain how this impacts the clinical decision-making process.

2. Broaden context:  

   Briefly compare findings with other cases or studies involving athletes to provide a broader perspective.

3. Streamline discussion:

   Reduce redundancy and focus on key points, particularly regarding imaging criteria and exercise-induced remodeling.

4. Highlight diagnostic challenges:  

   Add a short section emphasizing the clinical challenges of distinguishing physiological versus pathological LVHT and the implications for sports eligibility.

By addressing these minor concerns, the article will better align with its aim of providing clarity on LVHT in athletes.

Author Response

Thank you for valuable comments. Please find below our answers to your comments point-by-point.

R: 1. Clarify syncope pathophysiology: Clearly differentiate the role of LVHT from vasovagal syncope and explain how this impacts the clinical decision-making process.

Authors: we made attempts to clarify this topic inthe case description section and in discussions as well.

Case description

After a comprehensive review of the patient's medical history, lack of any registered arrhythmias during continuous telemonitoring of the patient and while performing exercise electrocardiography, it was considered that the patient experienced vasovagal syncope during marathon run. As a result, a tilt table test was performed, which revealed a mixed-type vasovagal syncope. Additionally, 24-hour Holter monitoring was performed and showed sinus rhythm with a heart rate (HR) of 28 – 91 beats per minute (bpm). Short bradycardia episodes with HR less than 25 bpm were observed. The average daily HR was 41 bpm.

Discussion

Athletes are generally healthy but might have certain cardiac disorders which might, during athletic participation or training, result in cardiac symptoms including syncope. Vasovagal syncope is probably the most common cause of syncope in athletes, but syncope in the context of these cardiac disorders might be a warning of sudden death. Consequently, the evaluation of syncope assumes extreme importance, particularly if these underlying cardiac conditions such as cardiomyopathy, myocarditis, channelopathies (long QT, Brugada syndromes etc.), pre-excitation syndrome, coronary artery anomalies or disease, are present (Baswaraj D, Flaker G. Syncope in Athletes: A Prelude to Sudden Cardiac Death? Mo Med. 2024 Jan-Feb;121(1):52-59. PMID: 38404441; PMCID: PMC10887456). In our patient the latter cardiac conditions have been excluded making vasovagal syncope the most probable reason for loss of consciousness during the marathon run.

R: 2. Broaden context: Briefly compare findings with other cases or studies involving athletes to provide a broader perspective.

Authors: this has been done in the discussion section. Please see all corrections highlighted in yellow and green.

R: 3. Streamline discussion:

Reduce redundancy and focus on key points, particularly regarding imaging criteria and exercise-induced remodeling.

Authors: this has been done in the discussion section. Please see all corrections highlighted in yellow and green. Important conclusions are present in the conclusion section which has been rewritten.

R:4. Highlight diagnostic challenges:

Add a short section emphasizing the clinical challenges of distinguishing physiological versus pathological LVHT and the implications for sports eligibility.

Authors: it has been clarified in discussion section (highlighted in green) and in conclusion section (it has been completely rewritten)

Reviewer 3 Report

Comments and Suggestions for Authors

The case is clinically relevant, addressing a diagnostic challenge in sports cardiology and emphasizing the importance of distinguishing physiological adaptations from pathological conditions. The manuscript is generally well-structured, clearly written, and supported by comprehensive imaging and follow-up data.

The introduction could better outline the diagnostic challenges and controversies in differentiating physiological from pathological LVHT, setting the stage for the case presentation. It would benefit from citing more specific prevalence rates for LVHT in various populations, especially athletes, to contextualize the case.

The case is well-detailed with clear timelines, diagnostic evaluations, and follow-up findings. The description of imaging findings (e.g., trabeculation measurements and fibrosis) is precise, but supplemental videos/figures should include annotations to guide readers unfamiliar with such imaging. The text mentions "various supplements" taken by the patient during adolescence but does not specify their nature. Clarifying this is important, as certain supplements can have cardiovascular implications. The management strategy could be elaborated, particularly on how exercise intensity was modified and monitored over the two-year follow-up.

The discussion would benefit from a deeper analysis of insertion point fibrosis. While it is described as benign, alternative interpretations and supporting references should be explored further.

The authors could expand on implications for broader clinical practice, such as the utility of specific diagnostic criteria in differentiating physiological LVHT from non-compaction cardiomyopathy.

The conclusions are concise but could more clearly emphasize actionable insights for clinicians managing athletes with suspected LVHT.

Author Response

Thank you for valuable comments. Please find below our anwers to your comments point-by-point.

R: The introduction could better outline the diagnostic challenges and controversies in differentiating physiological from pathological LVHT, setting the stage for the case presentation. It would benefit from citing more specific prevalence rates for LVHT in various populations, especially athletes, to contextualize the case.

Authors: new paragraph has been inserted to contextualize the case, highlighted in yellow (last two paragraphs in the introduction section).

R: The case is well-detailed with clear timelines, diagnostic evaluations, and follow-up findings. The description of imaging findings (e.g., trabeculation measurements and fibrosis) is precise, but supplemental videos/figures should include annotations to guide readers unfamiliar with such imaging.

Authors: Figure is annotated, it is impossible to annotate videos because of moving nature. In publications moving images are w/o annotations. Additional annotations has been added to the Figure 1.

R: The text mentions "various supplements" taken by the patient during adolescence but does not specify their nature. Clarifying this is important, as certain supplements can have cardiovascular implications.

Authors: it has been clarified: carnitine, amino acids (case presentation, 1st paragraph).

The management strategy could be elaborated, particularly on how exercise intensity was modified and monitored over the two-year follow-up.

Authors: it has been clarified stating that “patient restricted unsupervised physical activity to the mild to moderate non-competitive range” (case presentation section, 6th paragraph). Supervision was not performed.

R: The discussion would benefit from a deeper analysis of insertion point fibrosis. While it is described as benign, alternative interpretations and supporting references should be explored further.

Authors: Please find explanation in the text (discussion section, 8th paragraph)

R: The authors could expand on implications for broader clinical practice, such as the utility of specific diagnostic criteria in differentiating physiological LVHT from non-compaction cardiomyopathy.

Authors: it has been elaborated comprehensively in the discussion section (highlighted in green, Table 2).

R: The conclusions are concise but could more clearly emphasize actionable insights for clinicians managing athletes with suspected LVHT.

Authors: Conclusion section has been rewritten.

Round 2

Reviewer 1 Report

Comments and Suggestions for Authors

 Accept in present form.

Reviewer 3 Report

Comments and Suggestions for Authors

I would like to thank the authors for considering my comments and appropriately revising their manuscript